# The Impact of Anti-Inflammatory Drugs on the Prokaryotic Community Composition and Selected Bacterial Strains Based on Microcosm Experiments

**DOI:** 10.3390/microorganisms11061447

**Published:** 2023-05-30

**Authors:** Rózsa Farkas, Tamás Mireisz, Marwene Toumi, Gorkhmaz Abbaszade, Nóra Sztráda, Erika Tóth

**Affiliations:** 1Department of Microbiology, Doctoral School of Biology, Institute of Biology, Eötvös Loránd University, Pázmány Péter stny., 1/C, 1117 Budapest, Hungary; 2Department of Microbiology, Eötvös Loránd University, Pázmány Péter stny., 1/C, 1117 Budapest, Hungary; 3Department of Microbiology, Doctoral School of Environmental Sciences, Institute of Biology, Eötvös Loránd University, Pázmány Péter stny., 1/C, 1117 Budapest, Hungary

**Keywords:** microcosm experiments, aquatic habitat contamination, pharmaceuticals, biodegradation, microbial community composition

## Abstract

Non-steroidal anti-inflammatory drugs (NSAIDs) are increasingly recognized as potential environmental contaminants that may induce toxicity in aquatic ecosystems. This 3-week microcosm experiment explores the acute impacts of NSAIDs, including diclofenac (DCF), ibuprofen (IBU), and acetylsalicylic acid (ASA), on bacterial communities using a wide range of these substances (200–6000 ppm). The results showed that the NSAID-treated microcosms had higher cell count values than control samples, though the diversity of microbial communities decreased. The isolated heterotrophic bacteria mostly belonged to *Proteobacteria*, particularly *Klebsiella*. Next-generation sequencing (NGS) revealed that NSAIDs altered the structure of the bacterial community composition, with the proportion of *Proteobacteria* aligning with the selective cultivation results. Bacteria had higher resistance to IBU/ASA than to DCF. In DCF-treated microcosms, there has been a high reduction of the number of *Bacteroidetes*, whereas in the microcosms treated with IBU/ASA, they have remained abundant. The numbers of *Patescibacteria* and *Actinobacteria* have decreased across all NSAID-treated microcosms. *Verrucomicrobia* and *Planctomycetes* have tolerated all NSAIDs, even DCF. *Cyanobacteria* have also demonstrated tolerance to IBU/ASA treatment in the microcosms. The archaeal community structure was also impacted by the NSAID treatments, with *Thaumarchaeota* abundant in all microcosms, especially DCF-treated microcosms, while *Nanoarchaeota* is more typical of IBU/ASA-treated microcosms with lower NSAID concentrations. These results indicate that the presence of NSAIDs in aquatic environments could lead to changes in the composition of microbial communities.

## 1. Introduction

As a result of increasing drug consumption, a high number of pharmacologically active compounds (PhACs) are entering our natural water sources, posing an increasing ecological and health risk worldwide [1,2]. Today, the five most frequently used nonsteroidal anti-inflammatory drugs (NSAIDs) are acetylsalicylic acid, paracetamol, diclofenac, ibuprofen and naproxen [3]. These compounds have a negative effect on aquatic living organisms causing oxidative stress, DNA damage and behavioral changes in aquatic microscopic animals. Their photodegradation can produce compounds that are even more toxic than the original molecule [3,4,5,6]. To date, wastewater treatment plants (WWTPs) are not designed to entirely filter organic micropollutants, often characterized by high water solubility and poor biodegradability favoring their environmental persistence [7].

Many PhACs (e.g., antiepileptics, NSAIDs) have already been detected in aqueous environments, mainly in surface waters, groundwater, wastewater and, in some cases, in drinking water as well [8,9,10,11,12]. The Danube River, as one of the most important drinking water bases in Europe, is also endangered in this respect: the JDS3 (Joint Danube Survey 3) in 2013 [13] surveyed the entire length of the river and identified more than 100 different drug residues and other anthropogenic contaminants in it. Kondor et al. (2020) [14] also examined the Danube River and drinking water wells for drug residues; their experience showed that the gravel filtration bed of the Danube cannot remove chemical compounds completely.

One of the main concerns connected to the presence of pharmaceutical residues in the environment is their usual occurrence as complex mixtures rather than as a single compound. The different NSAIDs that are continuously and simultaneously used and released into the environment can interact synergistically [15,16]. Additionally, the environmental concentration of pharmaceutical compounds and their synergistic and antagonist effects are strongly related to the geographical area, climatologic conditions and the presence of wastewater discharges [17]. In the environment, the persistence of pharmaceutical products can be influenced by environmental factors, physicochemical properties of the molecule, molecular structure, polarity, photostability, chemical stability, volatility, presence of other pharmaceuticals in the same matrix and the presence and activity of microorganisms with the ability to degrade pharmaceuticals [18,19,20,21,22,23].

The ability of microorganisms to degrade drugs and their residues is becoming more important as it would be an economically and environmentally beneficial solution to increase organic micro-contamination. Microorganisms may be able to break down a number of difficult-to-degrade substrates (even polyaromatic compounds) facilitating their removal from the environment [24].

Numerous scientific papers have examined the impact of pharmaceuticals on bacteria across various taxonomic groups: Leão et al. [25] observed that the presence of salicylate increases *S. aureus* resistance to fluoroquinolones and fusidic acid, and it was also found [26] that this drug exhibits an increased resistance of *E. coli* to several antimicrobials (fluoroquinolones, ampicillin, cephalosporins, tetracycline and chloramphenicol).

Nevertheless, the environmental effects of drugs can extend far beyond the outcomes determined in toxicity assessments conducted on individual organisms—these substances have the potential to disrupt entire microbial communities. There are existing findings on the impact of drug residues on entire microbial communities (e.g., the finding of Gerbersdorf et al. [27]) but this research was found to be limited.

Based on a comprehensive review of the existing literature, no earlier studies have investigated the acute effects of NSAIDs in such a wide range and high concentrations as this research which aims to investigate the effect of three commonly used nonsteroidal anti-inflammatory drugs (NSAIDs): diclofenac (DCF), ibuprofen (IBU), and acetylsalicylic acid (ASA) on microbial communities in microcosm experiments originating water from the Danube River. This study combines multiple techniques such as (1) estimating cell count values by epifluorescent microscopy to reflect dynamic changes in the bacterial number; (2) selectively isolating bacteria utilizing NSAIDs as a sole carbon source; and (3) next-generation sequencing to further assess the impact of NSAIDs on microbial communities.

This study provides insights into the potential effect of NSAIDs on changes in microbial communities and highlights the importance of investigating their impact on natural water sources. The findings of this study may facilitate the development of novel bioremediation techniques to reduce the ecological and health risks posed by NSAIDs in natural water sources.

## 2. Materials and Methods

### 2.1. Experimental Design

Water samples have been taken two times from the right shore of the Danube in Budapest (Hungary, [47.477357° N; 19.061891° E]) into a 1.5 L clean, sterile, screw-capped glass bottle on 13 July 2020 (first experiment) and on 24 August 2020 (second experiment). The samples were transported to the laboratory at 4 °C temperature, and used for diclofenac (DCF), ibuprofen (IBU) and acetylsalicylic acid (ASA) exposure in laboratory microcosm experiments. An amount of 200 mL of water samples was dispensed into three types of microcosm systems (250 mL, clean, sterile glass bottles) separately loaded with the powdered form of the anti-inflammatory drugs to acquire 200, 500, 1000 and 2000 ppm concentration, respectively, regarding the first experiments (MK0_1, MK1, MD_200, MD_500, MD_1000, MD1_2000, MI_200, MI_500, MI_1000, MI1_2000, MA_200, MA_500, MA_1000, MA_2000), while 2000 and 6000 ppm in the case of the second experiments (MK0_2, MK2, MD2_2000, MD_6000, MI2_2000, MI_6000, MD_2000, MD_6000) (Table 1). Regarding both experiments, bottles were constantly shaking (80 rpm) for three weeks at 23 °C.

### 2.2. Determination of Physicochemical Parameters and Cell Count Values

Temperature, pH, conductivity and dissolved oxygen values were recorded during sampling on the field by a Hach HQ40D portable multimeter (Hach, Loveland, CO, USA). Determination of microscopic cell counts was carried out by using Nikon80i epifluorescent microscopy and NisElements program package [28]. For the investigations, 10 mL of water samples were filtered through isopore polycarbonate filters with a pore size of 0.22 µm (Millipore, Billerica, MA, USA) and fixed with 2% paraformaldehyde solution.

### 2.3. Selective Cultivation of Bacteria

Bacterial strains were isolated from MD_6000, MI_6000 and MA_6000 microcosms containing 6000 ppm of anti-inflammatory drugs. Isolation of strains was performed by standard dilution plate technique on DSM medium 1 supplemented with 2000 ppm of DCF, IBU and ASA, respectively. Incubation was carried out at 28 °C for 7 days. To better explore cultivable representatives utilizing the chosen drug residues, a 2nd selective isolation was carried out from separately created microcosms with 6000 ppm of drug residues based on the same experimental design, cultivation technique and laboratory circumstances as used previously.

### 2.4. DNA Extraction from the Bacterial Strains, 16S rRNA Gene Amplification and Taxonomic Identification

DNA was extracted from the isolated bacterial strains [29], then PCR amplification was done on the 16S rRNA gene using the primers 27F (5′–AGA GTT TGA TCM TGG CTC AG–3′) and 1492R (5′–GGT TAC CTT GTT ACG ACT T–3′) [30]. The 16S rRNA gene sequencing was carried out at LGC (Berlin, Germany). The sequenced bacterial strains were identified using EzBioCloud’s online identification system [31].

To analyze the genomic sequences and metabolic potential of selectively cultivated and identified bacterial strains, the genomic sequences of the type strains of these bacterial strains from the EzBioCloud database were obtained. By utilizing the Rapid Annotation using Subsystem Technology (RAST) pipeline [32], a web interface for functional annotation of genomic sequences to analyze the features and functions of these sequences, functions to the genes present in the genome based on their similarities to known genes in other organisms were assigned. Furthermore, the functional genes encoding the metabolism of aromatic intermediates in the bacterial strains were analyzed using the Kyoto Encyclopedia of Genes and Genomes (KEGG) database [33]. The genes responsible for the metabolism of these compounds in the bacterial strains were identified to gain insights into the potential metabolic capabilities of the strains.

### 2.5. DNA Extraction from Microcosms and Next-Generation DNA Sequencing

After incubation, the microcosms were filtered through a 0.22 μm pore size mixed cellulose filter (type GSWP; Millipore, Billerica, MA, USA). DNA was isolated from the filters using the Ultraclean^®^ PowerSoil DNA Isolation Kit (MoBio, Carlsbad, CA, USA) according to the manufacturer’s instructions.

For the identification of bacterial and archaeal community structure, the 16S rRNA genes of the DNA samples were amplified with PCR in separate, triplicate reactions using primers with the following target-specific sequences: Bact_341F (5′–CCT ACG GGN GGC WGC AG–3′) and modified Bact_805R (5′–GAC TAC NVG GGT ATC TAA TCC–3′) for Bacteria as well as A519F (5′–CAG CMG CCG CGG TAA–3′) and Arch855R (5′–TCC CCC GCC AAT TCC TTT AA–3′) for Archaea [34]. Before sequencing, the DNA concentration of the PCR products was determined using a Qubit meter (Invitrogen Life Technologies, Carlsbad, CA, USA), and a minimum concentration of 4 ng/μL and 50 μL of PCR product was respected. DNA sequencing was performed on the Illumina MiSeq platform using MiSeq standard v2 chemistry by the Genomics Core Facility RTSF (Michigan State University, East Lansing, MI, USA). The forward and reverse fastq files obtained from the Illumina sequencer were processed and analyzed using the Mothur v1.40.5 software [35]. The contigs were acquired using the make.contigs with a deltaq value of 10 to get sequences with high-quality scores. The screen.seqs command was applied to keep only the sequences having the expected length, number of polymers and ambiguous bases. The sequences were aligned to the Silva database (silva.nr_v132.align) [36], and the non-aligned sequences and columns containing only “.” were removed by using the screen.seqs and filter.seqs commands based on the position of the archaeal and bacterial primers within the 16S rRNA gene. The pre.cluster command was used to remove sequences with Illumina sequencing errors. The chimeric sequences were removed by the UCHIME algorithm [37]. Only the abundant sequences were kept using the command split.abund to split the sequences into two groups with a cutoff value equal to 1. The taxonomic classification of the sequences was done using the Silva database silva.nr_v132.tax, and the non-archaeal and non-bacterial sequences were removed from the analyses based on the taxonomic classification output. The OTUs (Operational Taxonomic Units) were calculated using a distance matrix with distances larger than 0.15 obtained by using the dist.seqs and later the cluster commands assign sequences to OTUs, and eventually, the consensus taxa were determined using the classify.otu. In the end, data were normalized using the sub.sample and then rarefaction.single and summary.single were applied to calculate the rarefaction curve data and the values of the diversity indices. Shannon–Weaver, inverse Simpson (1/D) diversity indices and Chao-1 metrics were calculated using Mothur [35].

Non-metric multidimensional scaling ordination (NMDS) of the microcosms with the Bray–Curtis dissimilarity index based on their bacterial and archaeal OTU composition was carried out with the PAST 4.03 program.

## 3. Results

### 3.1. Physicochemical Parameters of the Danube Water

The physicochemical characteristics of the sampling sites are summarized in Table 2.

### 3.2. Microscopic Cell Counts and Diversity Indices of the Samples

The microscopic cell counts are presented together with Shannon diversity indices in Figure 1 for both Archaea and Bacteria based on amplicon sequencing. The Shannon diversity indices were determined by analyzing the population diversity in the samples based on operational taxonomic units (OTUs). The index was calculated using the following formula: $H=-\sum_{i=1}^{s}\left(Pi\ln Pi\right)$ where *Pi* represents the proportion (n/N) of individuals of a specific species found (n) divided by the total number of individuals found (N), ln denotes the natural logarithm, Σ signifies the sum of the calculations, and *s* represents the number of species.

When analyzing both experiments, a difference could be observed between the absolute controls (before incubation) and the microcosms treated with NSAIDs (after incubation): after comparing MK0_1 and MK0_2 absolute controls to the first and second treated experiments, it was revealed that the cell count values of the treated experiments were higher by at least one order of magnitude. When comparing MK1 and MK2 controls (after incubation) to the first and second treated experiments, a significant difference was not found regarding the cell count values. The number of cells was the most outstanding in DCF-treated microcosms. After comparing the cell counts of both treated experiments, values were similar; even a high number of cells was observed in microcosms with 6000 ppm of drug residues. At the same time, when considering Shannon diversity indices of Bacteria, it was found that microcosms treated with lower concentrations (200 and 500 ppm) of NSAIDs showed higher diversity than the microcosms containing drug residues in high concentrations (1000, 2000 and 6000 ppm) the decreased cell count values did not go together with lower diversity indices. Among microcosms treated with NSAIDs, the ASA-treated microcosms showed the highest overall bacterial diversity while it was the lowest in DCF-treated ones. Diversity indices of Archaea were only 0 and 0.69, meaning that compared to Bacteria, archaeal diversity was not observed by next-generation sequencing in the microcosms.

### 3.3. Results of Selective Cultivation and Genomic Analysis of Bacterial Strains

Selective cultivation was performed from MD_6000 (containing 6000 ppm of DCF), MI_6000 (containing 6000 ppm of IBU) and MD_6000 (containing 6000 ppm of ASA) microcosms. The results of the taxonomic identification of our selectively cultivated strains are presented in Table 3. Results of the second round of selective cultivation from microcosms with 6000 ppm of DCF, IBU and ASA are presented in Table 4.

From MD_6000, altogether 3, from MI_6000, 8 and also from MA_6000, altogether 8 strains were cultivated on DSM 1 medium containing 2000 ppm of DCF, IBU and ASA. From the separately created microcosms with 6000 ppm of DCF, 25 and 6000 ppm of IBU, 16 strains were cultivated, but cultivation from the 2nd ASA microcosm was not successful. Based on the 16S rRNA gene sequence similarities of all the cultivated strains, they showed 98.54–99.91% similarity values to the reference sequences of the type strains.

All isolated heterotrophic bacteria belonged to *Proteobacteria* with the exception of *Paenibacillus lautus* belonging to *Firmicutes*. At the class level of *Proteobacteria*, the isolates belonged to *Alphaproteobacteria* (7 strains [12%]), *Betaproteobacteria* (7 strains [12%]) and *Gammaproteobacteria* (45 strains [76%]). Among *Alphaproteobacteria*, genera *Novosphingobium*, *Sphingobium* and *Ensifer* were found, while the betaproteobacterial representatives were mostly assigned to the genera *Burkholderia*, *Pseudoduganella* and *Cupriavidus*. Regarding *Gammaproteobacteria*, *Klebsiella*, *Pseudomonas*, *Aeromonas*, *Rahnella*, *Cytrobacter*, *Phytobacter*, *Raoultella* and *Kosakonia* were cultivated. Among them, *Klebsiella* was the most abundant bacterial genus.

The analyzed bacterial strains contained two outstanding genes responsible for beta-lactamase and fluoroquinolone resistance. Their occurrence was characteristic in almost all bacterial strains breaking down NSAIDs. Four genes were present in most of the bacterial species encoding the catechol and protocatechuate branches of beta-ketoadipate metabolism, the salicylate and gentisate metabolism, as well as the homogenization pathway for the breakdown of aromatic compounds. The beta-lactamase resistance gene was present in more than 90% of the bacteria while the fluoroquinolone resistance gene was found in all the studied species (Figure 2). In the case of bacteria degrading ASA and DCF, the catechol branch of the beta-ketoadipate pathway and the metabolism of salicylate and gentisate were encoded. Bacterial species degrading IBU, the gene encoding the homogenous pathway for the degradation of aromatic compounds and the protocatechol branch of the beta-ketoadipate pathway were the most common (Figure 3).

### 3.4. Results of Amplicon Sequencing

Bacterial sequences were more abundant than archaeal sequences in our amplicon libraries. NMDS ordination (Figure 4) of the microcosms with the Bray–Curtis dissimilarity index based on their bacterial and archaeal OTU composition confirmed that NSAIDs affect the bacterial and archaeal community structure of the microcosms; microcosms containing 2000 and 6000 ppm of NSAIDs are separated from the controls and treated microcosms with a lower concentration of drug residues.

#### 3.4.1. Bacterial Community Composition of the Microcosms Based on Amplicon Sequencing

Altogether, 36 bacterial phyla were identified by amplicon sequencing; the distribution of the most abundant members is presented in Figure 5a,b. Regarding MK0_1 and MK0_2 as the original samples (serving as absolute controls), the analysis revealed that *Proteobacteria* emerged as the dominant taxon, displaying the highest relative abundance. It was closely followed by *Bacteroidetes* and *Patescibacteria*, which exhibited significant representation within the samples. *Actinobacteria* were also found to be abundant in the analyzed samples. In contrast, *Verrucomicrobia*, *Cyanobacteria* and *Planctomycetes* displayed comparatively lower levels of representation. The taxonomic groups *Chlamydiae*, *Firmicutes* and *Dependentiae* exhibited even lower relative abundances within these samples.

The absolute dominance of *Proteobacteria* could be observed in all microcosms but their proportional ratio increases in the treated microcosms, especially in those treated with DCF: among *Alphaproteobacteria*, *Caulobacteraceae* (genus *Caulobacter*), *Sphingomonadaceae* and *Rhizobiales* (genus *Aquabacter* and genus *Sandarakinorhabdus*), regarding *Gammaproteobacteria*, *Burkholderiaceae* (genus *Polynucleobacter*), *Rhodocyclaceae* (genus *Aquicella*), *Enterobacteriaceae* (genus *Legionella* and *Acinetobacter*), as well as *Pseudomonadaceae* (genus *Pseudomonas*) were the most common representatives. The proportion of *Alpha*- and *Gammaproteobacteria* were similar in microcosms with 200–2000 ppm of NSAIDs, but the 6000 ppm of concentration was more favorable to *Gammaproteobacteria*. Among them, the two most abundant bacterial genera were *Pseudomonas* and *Acinetobacter*. In the absolute control samples, as already mentioned, the ratio of *Proteobacteria* was in balance with *Patescibacteria* and *Bacteroidetes*, especially in MK0_1, and then as the concentration of drugs was increased, *Proteobacteria* proliferated the treated microcosms. Their number was the most significant in the second experiment containing a high concentration of DCF (MD_6000), IBU (MI_6000) and ASA (MA_6000). When comparing the first and second microcosm experiments, it can be said that at 6000 ppm, *Proteobacteria* almost completely suppressed the other bacterial members (such as *Actinobacteria*, *Verrucomicrobia* or *Planctomycetes*) of the community except for *Bacteroidetes*, their abundance was still relatively high at this concentration. When comparing the DCF, IBU and ASA-treated microcosms with each other, it can be stated that *Proteobacteria* dominated the DCF-treated microcosms as they were the most abundant representatives in microcosms containing this anti-inflammatory drug. *Actinobacteria* (*Ilumatobacteraceae*, *Sporichthyaceae*, *Microbacteriaceae*) characterized the controls and microcosms treated with 200–500 ppm of IBU and ASA, were also abundant in MA2_2000 but low in DCF-treated microcosms. The ratio of *Unclassified Bacteria* was lower in DCF-treated microcosms than in the IBU- and ASA-treated ones. *Bacteroidetes* (*Chitinophagaceae*, *Saprospiraceae*, *Microscillaceae*, *Spirosomaceae*, *Ignavibacteria*) were revealed in all microcosms except MI1_2000, and they mostly dominated the IBU- and ASA-treated microcosms. Among *Chlamydiae*, *Chlamydiales*, *Parachlamydiaceae* and *Simkaniaceae* were present in the highest number, especially in MK0_1, MI_1000 and MA1_2000. *Cyanobacteria* were found in the controls and microcosms treated with lower, 200 ppm of IBU and ASA. Among *Dependentiae*, which ratio was high at 2000 ppm of ASA, *Babeliales* were mainly present. *Firmicutes* could only be found in the controls (*Bacillales*, *Clostridiales*, *Selenomonadales* and *Veillonellaceae*). As for *Patescibacteria*, consistently being observed in DCF-treated microcosms, *Parcubacteria* were the most abundant bacterial members. The number of *Planctomycetes* (*Phycisphaeraceae*, *Pirellulaceae*, *Rubinisphaeraceae*), similar to *Bacteroidetes*, was high in IBU- and ASA-treated microcosms. *Verrucomicrobia* (*Chthoniobacteraceae*, *Opitutaceae*, *Pedosphaeraceae*, *Verrucomicrobiaceae*) utilized DCF and IBU most successfully but were able to use lower concentrations (200–500 ppm) of ASA as well. Phyla contributing to less than 2% of the microcosms were *Acidobacteria*, *Armatimonadetes*, *Chloroflexi*, *Elusimicrobia*, *Entotheonellaeota*, *Epsilonbacteraeota*, *Fibrobacteres*, *Fusobacteria*, *Gemmatimonadetes*, *Hydrogenedentes*, *Kiritimatiellaeota*, *Latescibacteria*, *Lentisphaerae*, *Margulisbacteria*, *Nitrospirae*, *Omnitrophicaeota*, *Rokubacteria*, *Spirochaetes*, *Tenericutes* and *Zixibacteria*.

#### 3.4.2. Archaeal Community Composition of the Microcosms Based on Amplicon Sequencing

Altogether 8 archaeal phyla were identified by amplicon sequencing; the distribution of the most abundant members is presented in Figure 6. Regarding MK0_1 and MK0_2 as the original samples (serving as absolute controls), our analysis revealed that *Crenarchaeota* exhibited a minor representation, indicating a relatively lower abundance within the samples. Conversely, *Diapherotrites* and *Euryarchaeota* demonstrated a more pronounced presence, suggesting a higher abundance compared to *Crenarchaeota*. *Nanoarchaeaeota* displayed a substantial presence, indicating a significant abundance within the microbial community. *Thaumarchaeota* were also abundant in the samples, though they represented a relatively lower level compared to *Nanoarchaeota*.

Compared to the NSAID-treated microcosms, the archaeal communities of the control samples were more diverse. When analyzing the first and second experiments at the same time, it can be stated that the archaeal community has significantly changed in the microcosms, especially the treated ones: *Thaumarchaeota* (ammonia-oxidizing archaea [AOA]: *Nitrosopumilaceae*, *Nitrososphaeraceae* and *Nitrosotaleaceae*) were abundant in all microcosms but they dominated the microcosms treated with 200–2000 ppm of DCF and microcosms containing 6000 ppm of NSAIDs. *Nanoarchaeota* (*Nanohaloarchaeia* and *Woesearchaeia*) were also present in each microcosm, but they were more typical to IBU- and ASA-treated-microcosms with lower, 500–1000 ppm of concentrations. Further, *Thamarchaeota*, *Euryarchaeota* (*Thermoplasmata* and methanogenic archaeal representatives: *Methanobacterium*, *Methanobrevibacter*, *Methanocella*, *Methanocorpusculum*, *Methanoregula*, *Methanosaeta* and *Methanosarcina*) used high concentrations (6000 ppm) of drug-residues. When analyzing only the treated microcosms, it can be concluded that the archaeal composition of IBU- and ASA-treated microcosms were similar while the DCF-treated ones are differentiated from them mainly because of their lower *Nanoarchaeota* ratio. *Altiarchaeota* (*Altiarchaeia*) and *Crenarchaeota* (*Bathyarchaeia*) were typical to DCF-treated microcosms but the latter also could be found in MI1_2000. *Diapherotrites* (*Iainarchaeales* and *Micrarchaeia*) were observed in the controls and MD_1000. Unclassified Archaea were found in the controls, MD_500, MA_500 and MI2_2000.

## 4. Discussion

As mentioned before, a 3-week microcosm experiment was conducted in 2020 to assess microbial community changes influenced by exposure to anti-inflammatory drugs at various concentrations (200, 500, 1000, 2000 and 6000 ppm). NSAIDs trigger increasing toxic phenomena in aquatic environments. Based on JDS3 (2013), 20–40 ng L^−1^ of DCF, 550 ng L^−1^ of IBU and 470 ng L^−1^ ASA was measured in the Danube River. Several bacteria are able to utilize these compounds thus contributing to their biotransformation as well as degradation in the environment [24,25,26]. Our experiments had been focusing on microorganisms able to use NSAIDs as a sole carbon source rather than measuring their biodegradation capabilities. After cultivation, though, the genes of our isolated strains were analyzed upon NSAID metabolism and amplicon sequencing results were also compared to the literature data on the biodegradation possibility of NSAIDS.

In the microcosms, cell count values of the treated experiments were higher than the absolute controls, and then 6000 ppm of NSAID concentrations facilitated the proliferation of bacteria even more. When a community structure is interfered with in any way (e.g., it receives a selective inhibitor or some of the representatives are able to grow on a selective substrate), it always shifts the composition of the community and it is common that selective enrichment leads to a decreased diversity. The reduced diversity indices and the NGS analysis showed other interesting bacterial and archeal community changes that will be discussed below.

Among our cultivated bacterial strains, 1 belonged to *Firmicutes* while the rest of the strains were members of *Proteobacateria*. From the first isolation, representatives belonging to class *Alpha-*, *Beta-* and *Gammaproteobacteria* were found, while the second isolation resulted in strains found to be members only of *Gammaproteobacteria*. Regarding the first selective isolation, more taxa were cultivated than in the second; cultivation of representatives in the presence of ASA was also successful. Regarding the second selective cultivation, mostly genera *Aeromonas* and *Rahnella* were able to utilize DCF, while *Klebsiella* dominated the microcosms containing IBU. It is not surprising that mostly *Proteobacteria* were found since it was also revealed by the NGS analysis that the treatment of NSAIDs shifted the composition of the microbial communities in this direction.

The analyzed bacterial strains contained two outstanding genes responsible for beta-lactamase and fluoroquinolone resistance. Their occurrence was observable in almost all bacterial strains breaking down NSAIDs, but their presence hardly proves bacterial resistance to NSAIDs. Both fluoroquinolones and beta-lactam derivatives are widely used and distributed antibiotics. NSAIDs make microorganisms more sensitive to certain antibiotics, but regarding beta-lactam derivatives and fluoroquinolones, the opposite has been found, i.e., NSAIDs make bacteria more resistant to these compounds. This may happen since part of the bacterial resistance to these antibiotics is due to multidrug efflux pumps, which are substrates of NSAIDs and may be able to stimulate them. In addition, NSAIDs can alter the permeability of bacterial membranes [38,39].

Four genes were present in the bacterial species encoding the catechol and protocatechuate branches of beta-ketoadipate metabolism, the salicylate and gentisate metabolism, as well as the homogenization pathway for the breakdown of aromatic compounds. In this case, the drug molecule was the substrate of the efflux pump involved in the resistance against the given antibiotic or a gene encoding the breakdown of non-specific aromatic intermediates, which contributes to the resistance of the microorganisms to drug molecules as the aromatic ring is present in them [38,39,40,41,42].

By next-generation sequencing, it was found that the diversity of the treated microcosms was reduced compared to the controls, and except for *Proteobacteria*, a lower ratio of representatives could be observed in them. The type of pharmaceutical compound used for preparing the microcosms affected the selection of the microbial communities: when considering microcosms containing 200–2000 ppm of DCF, the abundance of *Proteobacteria* was high compared to microcosms with IBU and ASA in which a high number of *Bacteroidetes* and *Planctomycetes* could be observed in addition to *Proteobacteria*. At 6000 ppm, *Proteobacteria* almost completely overgrew other bacterial members except for *Bacteroidetes*.

*Proteobacteria* was described as the most diverse and representative phylum in marine and soil environments [43] dominating biological wastewater treatments due to the role of its high number of representatives in organic and nutrient removal [44]. It is also one of the most common phyla involved in the biodegradation of different pollutants, e.g., hydrocarbons [45] and pharmaceuticals [46]. A bacterial strain, *Labrys portucalensis* (*Alphaproteobacteria*), first was isolated from an industrially contaminated sediment in northern Portugal [47] and was able to biodegrade several fluoroanilines [48]. 4-Fluoroaniline is a common fluorine-containing aromatic amine, widely used in the synthesis of medicines and pesticides, such as the drug ezetimibe, norfloxacin and a 4-amidogen quinoline pesticide. In a study conducted by [49], *Labrys portucalensis F11* was shown to be able to biodegrade DCF. In another study, a considerable increase of *Alphaproteobacteria* was detected in activated sludge containing 500 and 1000 ppm of IBU [50], strongly correlating with our findings. *Sphingomonas* species can degrade salicylate and ibuprofen as well [51,52].

The two most abundant bacterial genera regarding *Gammaproteobacteria* were *Pseudomonas* and *Acinetobacter*. This finding is in correlation with a study examining the biodegradation of paroxetine (an antidepressant of the selective serotonin reuptake inhibitor [SSRI] class) and bezafibrate (a fibrate drug used as a lipid-lowering agent to treat hyperlipidemia) where these genera were also present in the highest number among *Gammaproteobacteria* [53]. Another finding also proved that the genus *Pseudomonas* is capable of aromatic compound degradation [24]. Research performed by Rutere et al. (2020) [54] found that *Pseudomonas thivervalensis* was able to grow by utilizing IBU as a sole carbon and energy source validating its ibuprofen degradation capabilities. Murdoch and Hay (2005) [52] summarized the metabolism of some compounds the most similar to ibuprofen and identified patterns in which *Pseudomonas putida* F1 and *Pseudomonas cepacia* had a role. *Pseudomonas putida* R1 was able to biodegrade salicylate, too [55]. It has been reported that bacteria belonging to *Methylococcaceae* could also be relevant to the metabolic degradation of IBU [56]. Interestingly, common aspects of the response of microcosms containing high, 2000–6000 ppm of IBU as well as gut microbiome response to IBU could be considered. As published, NSAIDs lead to changes in the composition and diversity of gut microbes in humans [57]. Even though the change in the gut microbial community is not directly connected to our study, the dominance of *Enterobacteriaceae* in the presence of a high concentration of IBU in both of these environments may suggest basic mechanisms that could be necessary to study further. Regarding the DCF biodegradation ability of the members of the family *Enterobacteriaceae*, it was also observed that an isolated bacterium *Enterobacter hormaechei* could degrade DCF at an elimination rate of 52.8%, while in the presence of glucose as a supplementary carbon source, the degradation rate increased to approximately 82% [58].

*Bacteroidetes* are heterotrophic bacteria, and some of them are able to degrade high molecular weight organic compounds like petroleum hydrocarbons [59]. The relatively higher percentage of *Bacteroidetes* in MD_6000, MI_6000 and MA_6000 might be explained by the increased concentration of NSAIDs. This could be supported by Drury et al. (2013) [60] who found that *Bacteroidetes* increased at an urban site with a higher concentration of complex organic compounds. Among *Bacteroidetes*, *Chitinophagaceae* was one of the most abundant families. Although a previous report was not found on their role in NSAID degradation, Kang et al. (2016) [61] published that *Chitinophagaceae bacterium* was capable of degrading 2-MIB, a T&O-causing compound.

The number of *Planctomycetes*, similar to *Bacteroidetes*, was high in IBU and ASA-treated microcosms except for MD_200 and was insignificant in the DCF-containing ones. De Bento Flores (2013) [62] described that DCF was the only studied chemical compound besides acetaminophen and caffeine that negatively affected the growth of *Planctomycetes*, causing the complete loss of cell viability. This effect of DCF could be in agreement with results reported by Yu et al. (2013) [63] about experiments with *Vibrio fischeri* and with our findings as well.

*Actinobacteria* characterized IBU- and ASA-treated microcosms (similarly to both *Bacteroidetes* and *Planctomycetes*) with 200–500 ppm of concentration, but could not use DCF. Compared to the controls, the number of *Actinobacteria* decreased but remained relatively high assuming adaption to the selected NSAIDs. *Actinobacteria* may degrade various aromatic compounds as well as biopolymers such as polyisoprenes (rubbers) [64]. According to Zhang et al. (2016) [65], the abundance of *Actinobacteria* increased in IBU-enriched planed beds. Pala-Ozkok et al. (2014) [66] also published that continuous exposure of activated sludge to erythromycin caused the predominance of members belonging to *Actinobacteria*. Another research study showed *Actinobacteria* could hydroxylate DCF to different products [67] with which our findings are not in harmony.

Members of *Verrucomicrobia* were present mainly in the DCF- and the IBU-treated microcosms but were also found in microcosms with lower concentrations (200–500 ppm) of ASA. In spite of that, this is one of the phyla abundant in microcosms treated with DCF, in addition to *Proteobacteria* and *Bacteroidetes*. Thelusmond et al. (2018) [68] observed that DCF negatively impacted *Verrucomicrobia* in DCF-amended soil samples. The first representatives of the phylum were first isolated as aerobic chemoorganotrophs from aquatic habitats [69], which is known to be related to several other phyla forming a cluster called a superphylum, including *Planctomycetes* and *Chlamydiae*, referring to them as the PVC superphylum (*Planctomycetes-Verrucomicrobia-Chlamydiae*) [70]. The group is little studied and has unknown scientific potential but often can be found in extreme environments [71]. Navrozidu et al. (2019) [4] reported that the proportion of *Verrucomibrobia* was high in the examined bacterial community when analyzed in immobilized biomass of a bioreactor fed with wastewater containing commercial ibuprofen tablets, but they were degraded after adding only pure ibuprofen to the samples.

The ratio of *Cyanobacteria* was the highest in 200 ppm of NSAID-containing microcosms, especially with IBU and ASA. Bácsi et al. (2016) [72] conducted an experiment similar to ours with Cyanobacteria: in addition to other NSAIDs, they also used DCF and IBU for testing and found that ibuprofen and piroxicam were less toxic than DCF, with which our study is in agreement.

Under extreme conditions (including the presence of toxic compounds), the most extremophilic phenotypes described so far belong to the Archaea domain [73], but upon their role, our knowledge is limited: most of the novelty discovered phyla do not have any cultured representatives [74], and are affiliated primarily with *Thaumarchaeota*, *Nanoarchaeota* and *Euryarchaeota*. They tended to increase in response to NSAID treatments, except for *Nanoarchaeota*, which seemed to decrease after raised concentrations of NSAIDs. Rutere et al. (2020) [54] reported similar mechanisms regarding the response of these archaeal phyla to IBU. According to Osorio et al. (2021) [75], biotransformation of DCF and its related NSAIDs can be microbially mediated by ammonia-oxidizing bacteria and archaea with which the high number of AOA *Thaumarchaeota* present in each of our NSAID-containing microcosms are in complete correlation. *Euryarchaeota* includes all methanogenic microorganisms [76]. Their presence in our microcosms can be connected to important metabolic pathways involved in the degradation of NSAIDs. Granatto et al. (2020) [77] also published that the abundant genus *Methanosaeta* was found in their examined anaerobic sludge containing DCF and IBU.

Overall, the bacterial strains isolated from microcosms enriched with anti-inflammatory residues indicate that these bacteria may have an active role in the degradation of these compounds. Cell count values did not decrease even at extremely high concentrations of NSAIDs, though the diversity of the communities decreased. The next-generation sequencing results showed that a few bacterial representatives (mainly *Proteobacteria* and *Bacteroidetes*) successfully proliferated during the test period in spite of high drug concentrations. The cultivation of NSAID-degrading bacteria allowed us to confirm that bacteria using drug residues as a sole carbon source can be resistant to different pharmaceutical pollutants. Further examinations should be performed to investigate the individual potential of our isolated strains degrading the selected pharmaceuticals as well as the findings by NGS to have the possibility to optimize the production of bacterial biomass for bioremediation applications and to contribute to various bioremediation procedures.

## 5. Conclusions

Despite the widespread use of NSAIDs, their unintended presence in the environment, the mechanisms of biological degradation and their genetic bases are becoming more and more understood. The need for isolation and characterization of bacterial strains being able to degrade anti-inflammatory residues is still important. The results of this work demonstrated that microorganisms are capable of adapting and responding to the presence of emergent pollutants, like pharmaceutical drugs. The isolated bacterial strains exhibited two genes associated with beta-lactamase and fluoroquinolone resistance. These genes were present in most bacterial strains capable of degrading NSAIDs, but their presence alone did not indicate resistance to NSAIDs. However, NSAIDs were found to make bacteria more resistant to beta-lactam derivatives and fluoroquinolones, potentially due to their effect on multidrug efflux pumps and bacterial membrane permeability. The taxonomic analysis revealed that the natural samples (serving as absolute controls) were dominated by *Proteobacteria*, with a high representation of *Bacteroidetes* and *Patescibacteria*. *Actinobacteria* were also found to be abundant, while *Verrucomicrobia*, *Cyanobacteria*, and *Planctomycetes* exhibited lower levels of representation. Regarding archaea, *Nanoarchaeaeota* were significantly present, indicating a substantial abundance within the microbial community, whereas *Thaumarchaeota*, although abundant, were relatively less represented compared to *Nanoarchaeota*. After three weeks of incubation, NSAIDs altered the structure of the microbial community of the samples, with an excessive proportion of *Proteobacteria*. Bacteria had higher resistance to IBU/ASA than DCF: in DCF-treated microcosms, there has been a high reduction in the number of *Bacteroidetes*, whereas in the microcosms treated with IBU/ASA, they have remained abundant. The numbers of *Patescibacteria* and *Actinobacteria* have decreased across all NSAID-treated microcosms. Verrucomicrobia and Planctomycetes have tolerated all NSAIDs, even DCF. *Cyanobacteria* have also demonstrated tolerance to IBU/ASA treatment in the microcosms. The changes in microbial communities in the presence of high concentrations of DCF, IBU and ASA suggest that environmental sites contaminated with these NSAIDs are likely to be recovered through bioremediation procedures. *Verrucomicrobia* may be the most important representative in the elimination of DCF, while *Bacteroidetes*, *Planctomycetes*, *Actinobacteria*, and even *Verrucomicribia* could all utilize IBU/ASA.

## Figures and Tables

**Figure 1 microorganisms-11-01447-f001:**
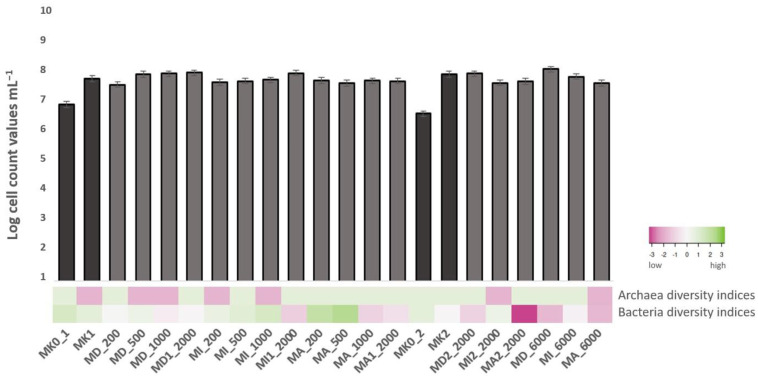
Cell count values with 0.1 standard error and Shannon–Wiener diversity indices of the microcosms before incubation (MK0_1; MK0_2) and after 3 weeks incubation (MK1; MD_200; MD_500; MD_1000; MD1_2000; MI_200; MI_500; MI_1000; MI1_2000; MA_200; MA_500; MA_1000; MA1_2000; MK2; MD2_2000; MI2_2000; MA2_2000; MD_6000; MA_6000).

**Figure 2 microorganisms-11-01447-f002:**
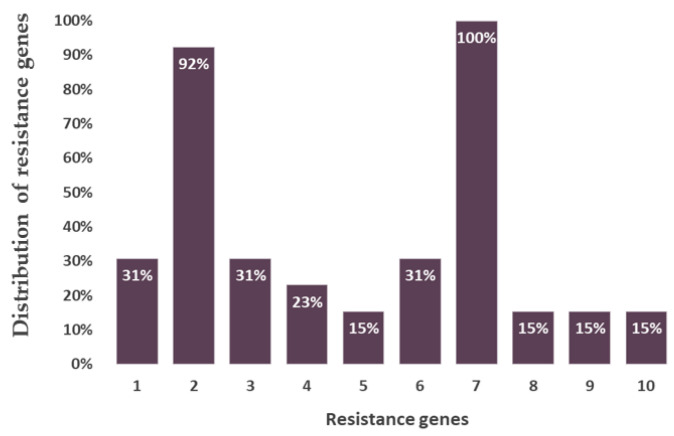
Distribution of antibiotic resistance genes in the studied, isolated bacterial strains. (1) Adaptation to D-cysteine; (2) beta-lactamase resistance; (3) bile hydrolysis; (4) phosphomycin resistance; (5) MexC-MexD-OprJ multidrug efflux system; (6) multidrug resistance efflux pumps; (7) fluoroquinolone resistance; (8) streptothricin resistance; (9) tetracycline resistance, ribosome protection type; (10) tetracycline resistance, ribosome protection type.

**Figure 3 microorganisms-11-01447-f003:**
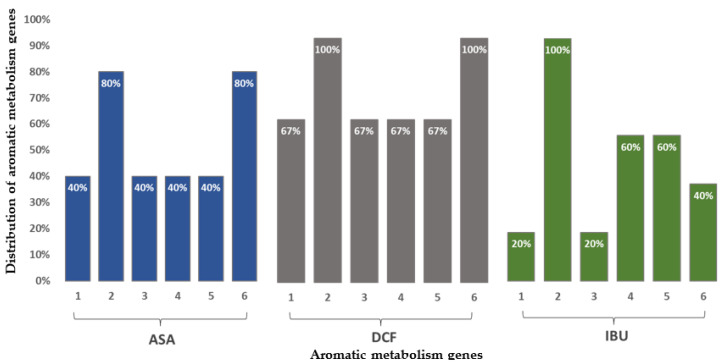
Distribution of aromatic metabolism genes in the studied, isolated bacterial strains degrading acetylsalicylic acid (ASA), diclofenac (DCF) and ibuprofen (IBU), respectively. (1) catabolic pathway of 4-hydroxyphenylacetic acid; (2) catechol branch of the beta-ketoadipate pathway; (3) central, meta-cleavage of aromatic compounds; (4) homogentisate pathway of catabolism of aromatic compounds; (5) protocatechol branch of the beta-ketoadipate pathway, (6): salicylate and gentisate.

**Figure 4 microorganisms-11-01447-f004:**
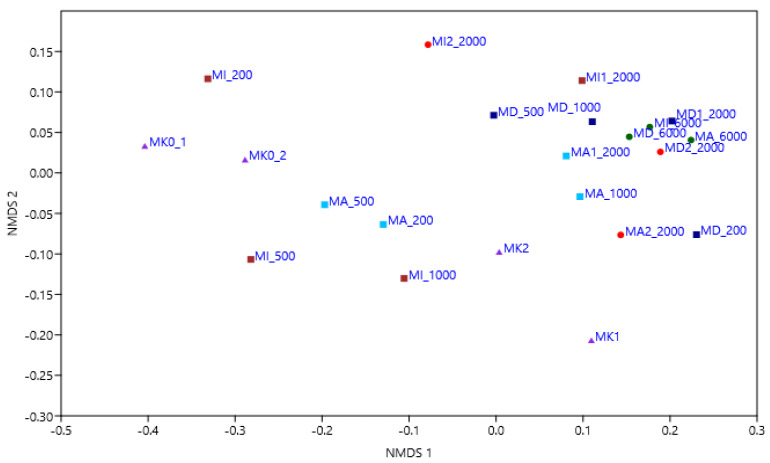
NMDS ordination of the microcosms with the Bray–Curtis dissimilarity index based on their bacterial and archaeal OTU composition (triangles: control samples, squares: first experiments (dark blue—with DCF, brown—with IBU, light blue—with ASA); dots: second experiments (red—2000 ppm of DCF, IBU and ASA; green: 6000 ppm of DCF, IBU and ASA)).

**Figure 5 microorganisms-11-01447-f005:**
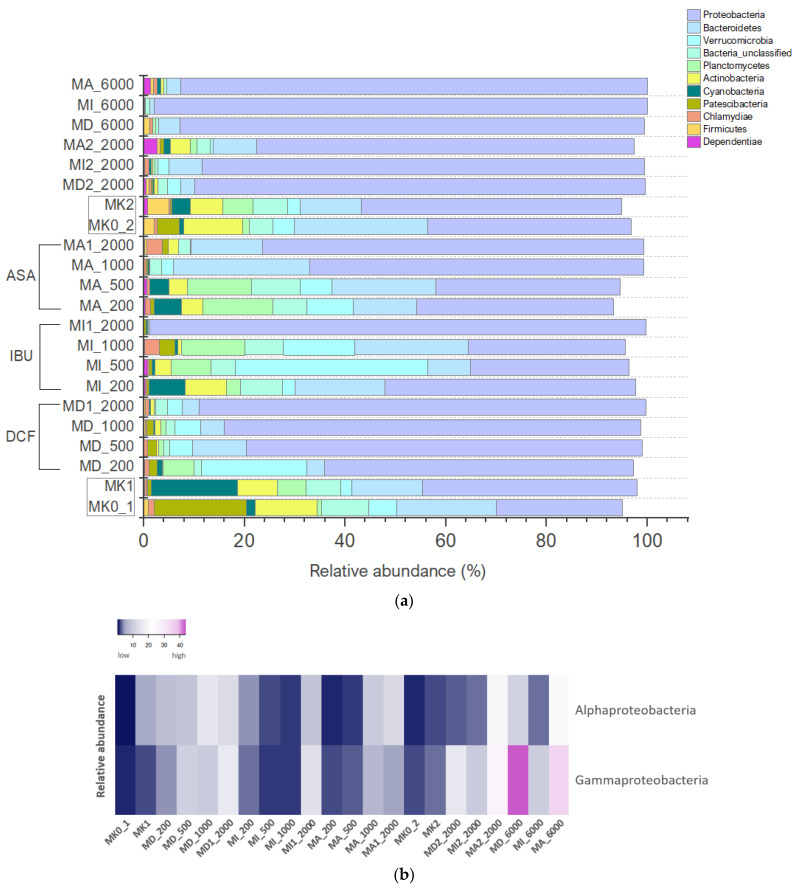
(**a**) Distribution of the abundant bacterial phyla based on 16S rRNA gene amplicon sequencing in the microcosms. (**b**) Distribution of the detected classes of the phylum Proteobacteria based on 16S rRNA gene amplicon sequencing in the microcosms.

**Figure 6 microorganisms-11-01447-f006:**
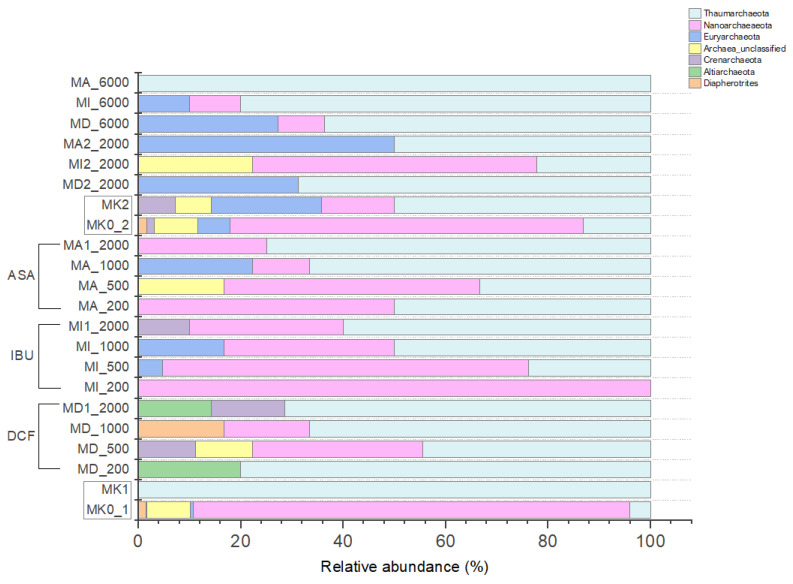
Distribution of the abundant archaeal phyla based on 16S rRNA gene amplicon sequencing in the microcosms.

**Table 1 microorganisms-11-01447-t001:** Name of the samples, control and first/second experiments of microcosms with anti-inflammatory drug content (ppm) and type (A: ASA [acetylsalicylic acid], D: DCF: [diclofenac], I: IBU [ibuprofen]).

Names of the Samples	Controls and Experiments	Anti-Inflammatory Drug Content (ppm)	Anti-Inflammatory Drug Applied
MK0_1	absolute control * (first)	0	none
MK1	control ** (first)	0	none
MD_200	first (July 2020)	200	DCF
MD_500	first (July 2020)	500	DCF
MD_1000	first (July 2020)	1000	DCF
MD1_2000	first (July 2020)	2000	DCF
MI_200	first (July 2020)	200	IBU
MI_500	first (July 2020)	500	IBU
MI_1000	first (July 2020)	1000	IBU
MI1_2000	first (July 2020)	2000	IBU
MA_200	first (July 2020)	200	ASA
MA_500	first (July 2020)	500	ASA
MA_1000	first (July 2020)	1000	ASA
MA1_2000	first (July 2020)	2000	ASA
MK0_2	absolute control * (second)	0	none
MK2	control ** (second)	0	none
MD2_2000	second (August 2020)	2000	DCF
MI2_2000	second (August 2020)	2000	IBU
MA2_2000	second (August 2020)	2000	ASA
MD_6000	second (August 2020)	6000	DCF
MI_6000	second (August 2020)	6000	IBU
MA_6000	second (August 2020)	6000	ASA

* absolute control: Danube water at the starting date of the experiments. ** control: the absolute control sample incubated for three weeks under the same conditions just like the treated microcosms, without NSAID.

**Table 2 microorganisms-11-01447-t002:** Physicochemical parameters of the sampling sites.

Experiments	Location	T °C	Conductivity (μS m^−1^)	pH	O_2_ (%)	O_2_ (mg L^−1^)
first (July 2020)	47.477357° N; 19.061891° E	17	223	7.83	95.9	12.54
second (August 2020)		18.5	203	7.57	93.6	13.22

**Table 3 microorganisms-11-01447-t003:** Results of the selective cultivation in the 1st cultivation experiment.

Strain Designation	Taxonomic Identification	Strain	Similarity (%)
MA1	*Burkholderia ambifaria*	AMMD	99.79
MA2	*Burkholderia ambifaria*	AMMD	99.81
MA4	*Novosphingobium arvoryzae*	JYI-02	100
MA6	*Paenibacillus lautus*	NBRC 15380	98.92
MA7	*Burkholderia vietnamiensis*	LMG 10929	99.89
MA8	*Burkholderia vietnamiensis*	LMG 10929	99.89
MA9	*Pseudoduganella danionis*	E3/2	98.54
MA10	*Burkholderia vietnamiensis*	LMG 10929	99.89
MD7	*Sphingobium yanoikuyae*	ATCC 51230	100
MD8	*Ensifer adhaerens*	Casida A	100
MD9	*Klebsiella grimontii*	06D021	100
MI1	*Pseudomonas nitritireducens*	WZBFD3-5A2	99.70
MI3	*Novosphingobium barchaimii*	LL02	99.43
MI4	*Klebsiella pneumoniae* subsp. *ozaenae*	ATCC 11296	99.90
MI10	*Sphingobium yanoikuyae*	ATCC 51230	100
MI11	*Novosphingobium barchaimii*	LL02	99.62
MI12	*Pseudomonas nitroreducens*	DSM 14399	99.77
MI13	*Sphingobium yanoikuyae*	ATCC 51230	100
MI14	*Cupriavidus agavae*	ASC-9842	98.70

**Table 4 microorganisms-11-01447-t004:** Results of the selective cultivation in the 2nd cultivation experiment.

Strain Designation	Taxonomic Identification	Strain	Similarity (%)
MD1	*Aeromonas sanarellii*	LMG 24682	99.90
MD2	*Aeromonas sanarellii*	LMG 24682	99.90
MD6	*Rahnella aceris*	SAP-19	98.49
MD7_2	*Rahnella aceris*	SAP-19	100.00
MD8_2	*Aeromonas caviae*	CECT 838	100.00
MD10	*Klebsiella michiganensis*	W14	98.92
MD12	*Aeromonas sanarellii*	LMG 24682	99.91
MD13	*Rahnella aceris*	SAP-19	100.00
MD15	*Aeromonas caviae*	CECT 838	98.95
MD17	*Rahnella aceris*	SAP-19	100.00
MD18	*Rahnella aceris*	SAP-19	100.00
MD19	*Klebsiella michiganensis*	W14	98.92
MD21	*Citrobacter bitternis*	SKKUI-TP7	99.81
MD22	*Aeromonas caviae*	CECT 838	98.95
MD23	*Rahnella aceris*	SAP-19	100.00
MD24	*Phytobacter diazotrophicus*	LS 8	99.39
MD25	*Rahnella aceris*	SAP-19	100.00
MD26	*Klebsiella grimontii*	06D021	100.00
MD27	*Klebsiella grimontii*	06D021	100.00
MD28	*Rahnella aceris*	SAP-19	100.00
MD29	*Aeromonas caviae*	CECT 838	98.95
MD30	*Aeromonas taiwanensis*	LMG 24683	99.50
MD31	*Aeromonas sanarellii*	LMG 24682	99.81
MD32	*Aeromonas sanarellii*	LMG 24682	99.90
MD33	*Klebsiella michiganensis*	W14	99.61
MI1_2	*Klebsiella quasivariicola*	KPN1705	99.80
MI2	*Klebsiella pneumoniae* subsp. *pneumoniae*	DSM 30104	99.61
MI3_2	*Pseudomonas aeruginosa*	JCM 5962	99.74
MI6	*Klebsiella huaxiensis*	WCHKl090001 ATCC 11296	99.84
MI9	*Klebsiella pneumoniae* subsp. *ozaenae*	ATCC 11296	99.90
MI10_2	*Klebsiella quasivariicola*	KPN1705	99.80
MI11_2	*Klebsiella quasivariicola*	KPN1705	99.80
MI12_2	*Klebsiella huaxiensis*	WCHKl090001	98.73
MI14_2	*Klebsiella huaxiensis*	WCHKl090001 W14	99.24
MI15	*Klebsiella michiganensis*	W14	99.65
MI17	*Raoultella ornithinolytica*	JCM 6096	100.00
MI18	*Klebsiella huaxiensis*	WCHKl090001	100.00
MI21	*Klebsiella michiganensis*	W14	99.44
MI22	*Kosakonia radicincitans*	DSM 16656	99.66
MI23	*Klebsiella huaxiensis*	WCHKl090001	100.00
MI24	*Klebsiella huaxiensis*	WCHKl090001	99.20

## Data Availability

The data that support the findings of this study are available on request from the corresponding author, [R.F.].

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
