# Peer review of "The Impact of Anti-Inflammatory Drugs on the Prokaryotic Community Composition and Selected Bacterial Strains Based on Microcosm Experiments"

_microorganisms, 2023, doi:10.3390/microorganisms11061447_

Round 1

Reviewer 1 Report

On account of the manuscript MICROORGANISMS-2336937, entitled “The impact of anti-inflammatory drugs on the prokaryotic community composition and selected bacterial strains based on microcosm experiments” by Rózsa Farkas et al., the authors evaluated the effect of three different anti-inflammatory drugs (NSAIDs) on microbial communities at different concentrations (200, 500, 1000, 2000, and 6000 ppm) based on a 3-week microcosm experiment. The topic is important to better understanding of the environmental impact of NSAIDs, and to conduct environmental management of NSAIDs in the aquatic environment as well. After careful consideration, I feel that this manuscript is to be published after improvement of some major shortcomings. Details of my comments are as follows:

1) The view point of this research is interesting, and the authors got interesting results. Several revisions are, however, required before publication. The present Abstract was not informative. Abstract should include purpose of the research, principal results and major conclusions in a summarized way. In addition, due to separation of the Abstract from the major article, it must be a key to lead readers to evoke a spirit of challenge to contact with the contents of the report. The authors are strongly encouraged to improve the Abstract for enhancement of the novelty and better understanding of the results.

2) Another aspect is in the novelty of the research. Although the authors mentioned the aim of this study, new aspect or view point of this research was not clearly stated in the manuscript. Introduction is not well structured. The authors don't necessarily mention general issues in detail, but are better to show information in a summarized way with focusing on the main issues related to the originality of this study. The authors are strongly encouraged to mention the new viewpoints and/or novel aspects which surpass the previous researches in the manuscript.

3) Experimental methodologies (validations) of distribution of resistance genes, and aromatic metabolism genes was missing in the present manuscript. In addition, quality assurance and/or quality control (QA/QC) for the confirmation of these analysis was not provided. The authors are encouraged to show these results more detail in the manuscript.

Reviewer 2 Report

The manuscript entitled Microbial Communities and Antimicrobial Resistance in Contaminated Aquatic and Terrestrial Environments is devoted to microcosm from the Danube River resistance to anti-inflammatory drugs (NSAIDs), diclofenac (DCF), ibuprofen (IBU) and acetylsalicylic acid (ASA). To date increasing drug consumption, and a high number of pharmacologically active compounds are entering our natural water sources, posing an increasing ecological and health risk worldwide. The ability of microorganisms to degrade drugs and their residues is becoming more important. The research topic is relevant and interesting, and this manuscript is consistent with the aims and scopes of the journal.

the introduction broadly describes the problem of river pollution with antibiotics and their components and contains the main necessary references to modern research.

There are several questions and remarks in the results section.

In the first section, it is worth describing in more detail the composition of natural samples and the content of antibiotics in them (perhaps Table 1 should be moved to the results)

In figure 1, what indices were used to determine the diversity of different groups?

In addition, in Figure 1, the data before and after incubation should be compared.

it is not clear whether a diversity analysis of the original samples has been carried out. It seems to me important to begin the description with the initial variety of water samples. perhaps this is the subject of section 3.7.1. where the analysis of microcosms is given. In this case, it should be moved to the beginning before the description of the isolates.

The conclusion should be extended and focused on comparing the content of antibiotics and their components in natural samples, the composition of the microbial community of natural samples, and the composition of isolate communities after 3 weeks of cultivation. in addition, it is important to identify which groups of microorganisms, according to the authors, are most important in the degradation of diclofenac (DCF), ibuprofen (IBU) and acetylsalicylic acid (ASA).

Reviewer 3 Report

Your manuscript focuses on the impact of three NSAIDs on structure of prokaryotic community. Overall, it is a well-organized and written paper.

Some aspects of the Ms. could/must be improve in order to facilitate its reading and comprehension. Please find bellow some suggestions/hints in that pursue.

General remarks

Improved the information in some figure captions

Edit some orthographic mistakes

- Abstract: Correct the meaning of NSAIDs, as you have in “Introduction”

- line 124 “Whole genome sequences of the type strains of our selectively cultivated and identified bacterial strains were downloaded also from the database of EzBioCloud to analyze their genome features and functions by RAST (Rapid Annotation using Subsystem Technology), a web interfaced pipeline [31]”. – why use the type strains and not your isolates? Explain the focuses on resistance genes against antibiotics for your goal.

- Paragraph (lines 169 to 173) seems repeating the same information within lines 124-128.

- line 175 - 3.1. Physicochemical parameters of the Danube water, Physicochemical characteristics of the sampling sites are summarized in Table 2. – I think this point should be in the material and methods section, explaining the origin of the water samples.

Results

- Figure 1 – improve the caption. The significance of all acronyms MK0-1, (…), etc., must be explained, as well the error bar in each column (standard-deviation, standard-error? n=?) – this suggestion is extended to other figure and table captions (e.g. Tables 4 and 5, Fig 4, Fig 5, Fig.6)

- Do the statistics for comparing substances or/and concentrations in terms of diversity indexes and cell counts.

- Line 185/186 – “significant difference was not found regarding the cell count values” – what statistical test? By “significant” you mean?

- Please give a space between table 5 and the line 211.

- Figure 2 – what isolated bacterial strains? The ones in Table 4 and 5? Same observation for Fig.3.

- Fig. 4 – the cycle within has some significance?

- 3.7.1. Bacterial community composition of the microcosms based on amplicon sequencing – I suggest the use of a Venn diagram for a better visualization and differenciation of the bacterial communities among the drugs/concentrations.

Discussion:

Lines 374/375 - “This may happen 374 since part of the bacterial resistance to these antibiotics is due to multidrug efflux pumps, 375 which are substrates of NSAIDs, and may be able to stimulate them” – could you explain better?

Round 2

Reviewer 1 Report

On account of the manuscript MICROORGANISMS-2336937R1, entitled “The impact of anti-inflammatory drugs on the prokaryotic community composition and selected bacterial strains based on microcosm experiments” by Rózsa Farkas et al., the author revised the manuscript appropriately according to the Reviewers comments. After careful consideration, I made a decision that the manuscript is acceptable for publication in its present form.

Reviewer 2 Report

the authors have corrected the text in accordance with all my comments